# ABXI: Invariant Interest Adaptation for Task-Guided Cross-Domain Sequential Recommendation

Submission Id: 227

## ABSTRACT

Cross-Domain Sequential Recommendation (CDSR) has recently gained attention for countering data sparsity by transferring knowledge across domains. A common approach merges domain-specific sequences into cross-domain sequences, serving as bridges that enable mutual enhancement between domains. One key challenge is to correctly extract the effective shared knowledge among these sequences and appropriately transfer it. Most existing works directly transfer unfiltered cross-domain knowledge rather than extracting domain-invariant components and adaptively integrating them into domain-specific modelings. Another challenge lies in aligning the domain-specific and cross-domain sequences. Existing methods align these sequences based on timestamps, but this approach can cause prediction mismatches when the current tokens and their targets belong to different domains. In such cases, the domain-specific knowledge carried by the current tokens may degrade performance. To address these challenges, we propose the A-B-Cross-to-Invariant Learning Recommender (**ABXI**). Specifically, leveraging LoRA's effectiveness for efficient adaptation as supported by numerous studies, our model incorporates two types of LoRAs to facilitate the adaptation process. First, all sequences are processed through a shared encoder that employs a domain LoRA for each sequence, thereby preserving unique domain characteristics. Next, we introduce an invariant projector that extracts domain-invariant interests from cross-domain representations, utilizing an invariant LoRA as well to adapt these interests into recommendations in each specific domain. Besides, to avoid prediction mismatches, all domain-specific sequences are re-aligned to match the domains of the cross-domain ground truths. Experimental results on three datasets demonstrate that our approach achieves better results than other CDSR counterparts, with an average improvement of 17.30% in HR@10 and 18.65% in NDCG@10. The codes are available in https://anonymous.4open.science/status/ABXI-WWW25-1D04.

## CCS CONCEPTS

• **Information systems** → **Recommender systems**.

## KEYWORDS

Recommender Systems, Cross-Domain Sequential Recommendation, Low-Rank Adaptation

*Conference acronym 'XX, June 03–05, 2018, Woodstock, NY*
© 2024 Association for Computing Machinery.
ACM ISBN 978-1-4503-XXXX-X/18/06…$15.00
https://doi.org/10.1145/nnnnnnn.nnnnnnn

**ACM Reference Format:**
Anonymous Author(s). 2024. ABXI: Invariant Interest Adaptation for Task-Guided Cross-Domain Sequential Recommendation. In *Woodstock '18: ACM Symposium on Neural Gaze Detection, June 03–05, 2018, Woodstock, NY*. ACM, New York, NY, USA, 10 pages. https://doi.org/10.1145/nnnnnnn.nnnnnnn

## 1 INTRODUCTION

In the era of information explosion, the Internet is flooded with massive amounts of content, yet users are exposed to only a small fraction of it. Such data sparsity remains a persistent challenge in modern recommender systems. Cross-Domain Sequential Recommendation (CDSR) has recently emerged as a promising approach to alleviate this sparsity issue by transferring knowledge across different domains to enrich user profiles [4, 5, 9, 11, 30, 41, 53, 55, 60].

A common strategy in CDSR involves merging domain-specific sequences into cross-domain sequences that serve as bridges, enabling mutual enhancement between domains [4, 9, 30, 53, 55]. Figure 1 illustrates an example of a user's domain-specific and cross-domain interaction sequences. In the book domain, the user's interests encompass science fiction and romantic novels, while in the movie domain, the user prefers science fiction and comedy films. From the perspective of cross-domain sequences, the user's interest in science fiction can be leveraged in both the book and movie domains to create more comprehensive user profiles. On the contrary, the specific interests in romantic books and comedy movies should not be indiscriminately shared between domains. However, most existing CDSR approaches mix up the concepts of cross-domain and domain-invariant interests by directly transferring unfiltered cross-domain knowledge into domain-specific modeling. This practice can introduce domain-specific information from one domain to interfering with another, adversely affecting recommendation performance. Therefore, extracting domain-invariant knowledge from cross-domain sequences is essential to facilitate effective sharing across specific domains.

Furthermore, another challenge lies in aligning domain-specific and cross-domain sequences when making recommendations within each domain. Current self-attention-based methods typically align cross-domain and domain-specific sequences based on timestamps [9, 30, 55], as depicted in Figure 2b. Although this approach is intuitive and facilitates the enhancement of cross-domain sequential features with domain-specific features through direct token-wise addition, it has inherent limitations. Generally, cross-domain training sequences comprise a mixture of items from different domains. If the current token and its ground truth token belong to different domains, the domain-specific information encoded in the current token may negatively impact the prediction of the ground truth token. For instance, as illustrated in Figure 2b, consider selecting token A4 as the current token to predict token B4. The timestamp-guided alignment enables the model to incorporate encoded cross-domain interests along $seq_X$ as well as encoded domain-specific interests

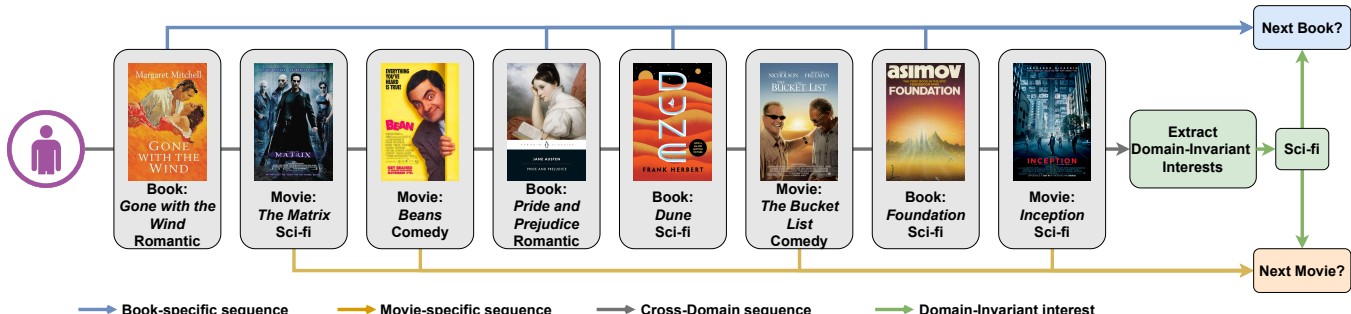

**Figure 1: Our proposal on generating recommendations by integrating domain-specific interests with domain-invariant interests extracted from the cross-domain sequence.**

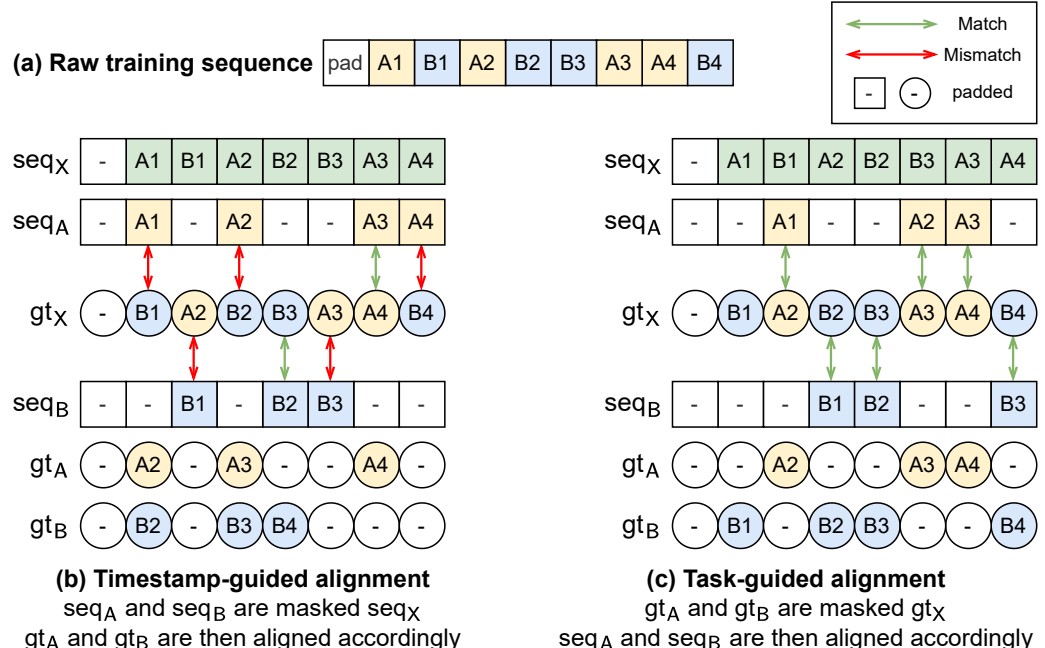

**Figure 2: Illustration of the sequence splits under different alignments, where gt denotes the ground truth. (a) illustrates the input training sequence. (b) and (c) demonstrate the split outcomes of timestamp-guided and task-guided alignment, respectively.**

along $seq_A$. However, since the target token B4 originates from domain B, it does not correspond with the A-specific knowledge, potentially degrading the model's performance. We refer to this issue as the prediction mismatch throughout the remainder of this paper.

To tackle the prediction mismatch issue and address the challenges of exploiting domain-invariant interest, we propose the A-B-Cross-to-Invariant Learning Recommender (**ABXI**). Specifically, we first realign all domain-specific sequences according to the domains of the ground truths to prevent prediction mismatches with cross-domain sequences. We then employ a shared self-attention encoder as the sequence model to encode all sequences into sequential representations. This shared encoder deploys one domain LoRA (**dLoRA**)

for each sequence, which can efficiently switch modes to encode every cross-domain and domain-specific sequence. Additionally, we instantiate an invariant projector to extract the domain-invariant interests from cross-domain representations. This projector has integrated one invariant LoRA (**iLoRA**) for recommendations in each specific domain to conduct efficient adaptation. While LoRAs are typically used for fine-tuning, we extend their application to single-stage training by concurrently training LoRA modules with all other components in ABXI. Having introduced these designs, ABXI renovates both the pipelines of obtaining cross-domain and domain-specific interests.

To thoroughly evaluate ABXI with state-of-the-art CDSR methods, we conduct extensive experiments on three publicly available

datasets. Experimental results show that ABXI outperforms all baselines by a significant margin, achieving notable improvements of 17.30% on HR@10 and 18.65% on NDCG@10. Ablation studies and sensitivity analyses further demonstrate the effectiveness of our proposed designs.

To conclude, our contributions can be summarized as follows:

- We identify the prediction mismatch problem within previous sequence-model-based CDSR works, and introduce a task-guided alignment to solve this problem.
- We introduce two types of LoRA: dLoRAs switch the mode of the encoder to handle encoding each sequence; iLoRAs adaptively integrate domain-invariant interests into recommendations in each specific domain.
- Extensive experiments are provided to demonstrate the effectiveness of ABXI. Results show that ABXI outperforms all baselines including state-of-the-art CDSR counterparts.

The rest of this paper is organized as follows: Section 2 provides an overview of related work. In Section 3, we formalize the CDSR problem we aim to solve and introduce our proposed ABXI. Section 4 evaluates ABXI through extensive experiments; Section 5 presents the conclusion.

## 2 RELATED WORK

### 2.1 Cross-Domain Recommendation

Cross-Domain Recommendation (CDR) leverages transfer learning techniques to mitigate data sparsity. Common methods include domain alignment, which aligns users' or items' representations across different domains [36, 50, 58], and domain adaptation, which adapts source knowledge to enhance target domains [10, 16, 27, 33, 61]. Besides these typical CDR works, Multi-Modal Recommendation (MMR) can also be considered a form of CDR, as different modalities can be viewed as domains due to their shared semantics [18, 28, 51, 52].

Recently, Large Language Models (LLMs) have attracted much attention for their strong performance and scalability [2, 47]. Researchers have attempted to introduce LLMs into recommender systems as well [1, 31]. However, these LLMs are Pretrained Language Models (PLMs) that are pretrained on Natural Language Processing (NLP) tasks. Therefore, researchers need to adapt these models to the recommendation domain. Under this perspective, using PLMs for recommendation can be seen as a cross-domain approach, where the source domain is the pretrained NLP domain, and the target domain is the recommendation domain.

A widely used solution is to leverage Parameter-Efficient Fine-Tuning (PEFT) techniques to perform this adaptation at affordable costs. Most works utilize Low-Rank Adaptation (LoRA) [15] for such adaptation [1, 29, 31, 56, 59]. Other techniques are also employed, such as prompt tuning [23, 25, 42, 44, 54, 57] and adapter tuning [12, 17].

### 2.2 Cross-Domain Sequential Recommendation

Sequential Recommendation (SR) aims to predict users' next interacted items based on their historical interaction sequences [7, 21, 45]. Similar to CDR, CDSR introduces transfer learning into SR to conduct knowledge transfer in sequential scenarios. Early CDSR

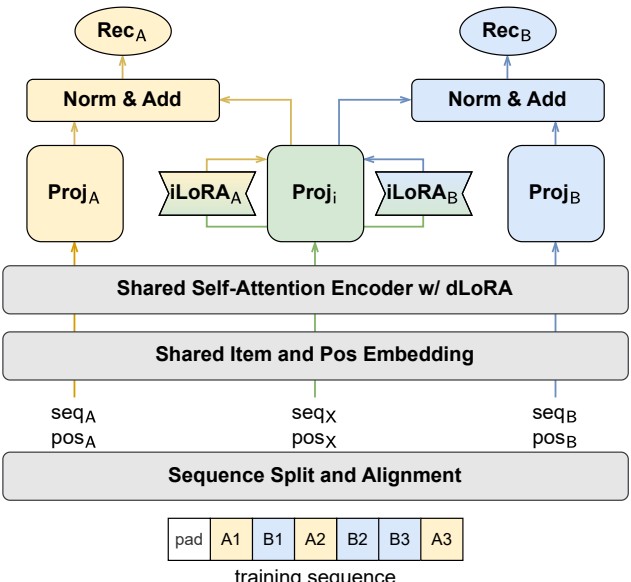

**Figure 3: Processed ABXI model.**

works focus on the assumption of multiple users sharing the same account [14, 38, 39, 46].

More recently, a broader concept of Cross-Domain Sequential Recommendation (CDSR) has emerged, focusing on leveraging bridging knowledge to enhance performance across domains. Depending on the type of knowledge transferred, CDSR approaches can be categorized into several types. Some studies [24, 26, 30] leverage overlapping users who have interacted in both domains as bridges to enhance performance for all users, including non-overlapping ones. Some methods [4, 5, 9, 53, 55] focus exclusively on overlapping users to strengthen their profiling in target domains. IESRec [35] leverages semantic similarities in natural language to align domains in scenarios without overlapping users. MAN [30] utilizes user groups to facilitate aligning domains. Additionally, similar to MMR, researchers have introduced multi-modal data into Sequential Recommendation (SR) to develop Multi-Modal Sequential Recommendation (MMSR) models [8, 20, 49].

## 3 METHODOLOGY

### 3.1 Problem Formulation

In this study, we focus on the dual-target CDSR task, involving two distinct domains denoted as A and B. Let a user's historical interaction sequences in domains A and B be represented as $seq_A = (i_{A1}, i_{A2}, i_{A3}, \ldots, i_{An})$ and $seq_B = (i_{B1}, i_{B2}, i_{B3}, \ldots, i_{Bm})$, respectively. The objective is to predict the user's next interaction items in each domain, specifically $i_{A(n+1)}$ and $i_{B(m+1)}$. The task can be formulated as:

**Input**: One user's domain-specific sequences, $seq_A = (i_{A1}, i_{A2}, i_{A3}, \ldots, i_{An})$ and $seq_B = (i_{B1}, i_{B2}, i_{B3}, \ldots, i_{Bm})$.

**Output**: A recommender system that estimates the probability of this users' next items, $i_{A(n+1)}$ and $i_{B(m+1)}$, to interact.

## 3.2 Overview

The architecture of our proposed ABXI model, as depicted in Figure 3, comprises four components: (1) sequence formulation and embedding; (2) shared self-attention encoder with domain LoRA; (3) projectors with invariant LoRA; and (4) the optimization objective.

## 3.3 Sequence Formulation and Embedding

Given a user's raw training sequence, we first extract the last interaction and the first interaction to form the cross-domain sequence $seq_X$ and the cross-domain ground truth $gt_X$, following the seq2seq paradigm. For simplicity, we denote the combined domain as domain X. As illustrated in Figure 2c, we then derive the domain-specific ground truths $gt_A$ and $gt_B$ by masking the respective domains in $gt_X$.

Subsequently, we create the domain-specific sequence $seq_A$ and $seq_B$ by masking respective domain in $seq_X$ and re-aligning them based on $gt_A$ and $gt_B$ with paddings. The aligned sequences $seq_A$ and $seq_B$ are ensured to have the same target items, position-wise, as $seq_X$. In this way, the prediction mismatch issue encountered by previous works [9, 55] is addressed.

Besides, we use $gt_A$ and $gt_B$ solely as intermediate terms to re-align domain-specific sequences, and they are not utilized during optimization. Consequently, the ground truths of $seq_X$ and its $seq_A$ and $seq_B$ are unified, which eliminates the need for standalone domain-specific recommendation loss. The position indices of each sequence are assigned separately in reverse chronological order.

Each item is then embedded into a learnable vector at length $d$. We initialize the item embedding table $\mathbf{E}_I \in \mathbb{R}^{N \times d}$, where $N$ the total number of items, and the position embedding table $\mathbf{E}_P \in \mathbb{R}^{L \times d}$, where $L$ is the maximum sequence length. Besides that, we also initialize the position embedding table $\mathbf{E}_P \in \mathbb{R}^{N \times d}$. Both embedding tables are shared across all sequences. Finally, The sequence embeddings for each sequence are obtained by adding the item and position embeddings, followed by a dropout operation to mitigate overfitting. We denote these sequence embeddings as $E_X$, $E_A$ and $E_B$ for domain X, A and B, respectively.

## 3.4 Low-Rank Adaptation

Many recommendation system studies [13, 19, 22, 37] have already demonstrated that index-based methods exhibit low-rank natures because of data sparsity. LoRA [15], designed as a PEFT technique, exploits similar low-rank characteristics in data to conduct efficient task adaptations [3]. We leverage LoRA to conduct efficient domain adaptation in recommendation by proposing two modules: domain LoRA (**dLoRA**), which helps shared encoders adapt to both cross-domain and domain-specific modeling; and invariant LoRA (**iLoRA**), which adapts the extracted domain-invariant knowledge into specific final recommendations. Besides, notice that incorporating dropout operators can mitigate overfitting in LoRAs [32]; Therefore, the forward pass of our proposed LoRAs yields:

$$\text{LoRA}(X) = \text{Drop}\left(M_B^\uparrow M_A^\downarrow X\right), \quad (1)$$

where Drop depicts dropout operator, $M_A^\downarrow \in \mathbb{R}^{r \times d}$, and $M_B^\uparrow \in \mathbb{R}^{d \times r}$ denote the down- and up-projection matrices with rank $r < d$. We argue that LoRA's potential is not confined to multi-stage training

but can be effectively applied in single-stage training as well. Thus, all LoRAs in ABXI are trained together with the rest of the model.

## 3.5 Shared Encoder with Domain LoRAs

Inspired by the effectiveness of SASRec [21], numerous sequential recommenders adopt self-attention encoder as the backbone sequence model [6, 43, 45]. Among them, all CDSR models instantiate multiple self-attention encoders for modeling sequences from different domains [4, 5, 9, 30, 55].

We posit that a single self-attention encoder is sufficient to capture the majority of the necessary knowledge for recommendations, given the overlap in domain-invariant knowledge across domains. Consequently, we instantiate one shared encoder to all sequences. To preserve the specific uniqueness of each domain, we introduce three dLoRA modules in parallel with the encoder's feedforward network. These dLoRAs enable the shared encoder to switch modes efficiently among the domains X, A, and B, thereby maintaining domain-specific nuances without compromising shared knowledge. The encoding process for a domain-X sequence embedding $E_X$ is formulated as follows:

$$H_X = \text{LN}\left(E_X + \text{Drop}\left(\text{MHA}\left(E_X\right)\right)\right), \quad (2)$$

$$H_X^{\text{enc}} = \text{LN}\left(H_X + \text{Drop}\left(\text{FFN}\left(H_X\right)\right) + \text{dLoRA}_X\left(H_X\right)\right), \quad (3)$$

where LN denotes LayerNorm, $\text{dLoRA}_X$ denotes the dLoRA unit for domain X, MHA denotes the multi-head attention networks, and FFN denotes the feedforward networks. Similarly, by replacing the domain notation X with A and B in Eqs. 2 and 3, we obtain the encoded sequential representations $H_A^{\text{enc}}$ and $H_B^{\text{enc}}$, respectively.

## 3.6 Projectors with Invariant LoRAs

To convert the encoded sequential representations into effective recommendation representations, we utilize a dedicated projector for each domain. Each domain-specific projector consists of a SwishGLU variant of MLP [47]. The structure of a projector is defined as follows:

$$\text{Proj}\left(X\right) = \text{Drop}\left(\left(\text{Swish}\left(XW_1\right) \otimes H_A^{\text{enc}}W_2\right)W_3\right), \quad (4)$$

where $W_1 \in \mathbb{R}^{d \times \frac{8}{3}d}$, $W_2 \in \mathbb{R}^{d \times \frac{8}{3}d}$ and $W_3 \in \mathbb{R}^{\frac{8}{3}d \times d}$ are learnable matrices. The Swish activation function is defined as $\text{Swish}(x) = \frac{x}{1+e^{-\beta x}}$, with $\beta$ set to 1 [47].

For projectors within specific domains, we incorporate skip connections to obtain the projected domain-specific representations:

$$H_A^{\text{p}} = \text{LN}\left(H_A^{\text{enc}} + \text{Proj}_A\left(H_A^{\text{enc}}\right)\right), \quad (5)$$

$$H_B^{\text{p}} = \text{LN}\left(H_B^{\text{enc}} + \text{Proj}_B\left(H_B^{\text{enc}}\right)\right). \quad (6)$$

In contrast to domain-specific projectors, the invariant projector integrates two iLoRAs to adapt domain-invariant interests into final recommendations for domains A and B. The projected invariant representations are obtained as follows:

$$H_{\text{i2A}}^{\text{p}} = \text{LN}\left(H_X^{\text{enc}} + \text{Proj}_i\left(H_X^{\text{enc}}\right) + \text{iLoRA}_A\left(H_X^{\text{enc}}\right)\right), \quad (7)$$

$$H_{\text{i2B}}^{\text{p}} = \text{LN}\left(H_X^{\text{enc}} + \text{Proj}_i\left(H_X^{\text{enc}}\right) + \text{iLoRA}_B\left(H_X^{\text{enc}}\right)\right). \quad (8)$$

Here, $\text{Proj}_i$ shares the same structure as $\text{Proj}_A$ and $\text{Proj}_B$, while $\text{iLoRA}_A$ and $\text{iLoRA}_B$ are the two instantiations of iLoRA in ABXI.

The final recommendation representations are obtained by summing the projected invariant and domain-specific representations:

$$H_A^{rec} = H_A^p + H_{i2A}^p, \qquad (9)$$

$$H_B^{rec} = H_B^p + H_{i2B}^p. \qquad (10)$$

## 3.7 Optimization Objective

Previous works' adopting timestamp-guided alignment [9, 55] requires separate optimization on the cross-domain sequences. However, since our proposed task-guided alignment unites the ground truths of domain-specific and cross-domain sequences, ABXI can be optimized entirely with only one set of positive and corresponding negative samples. We split the input training sequences in the manner of the seq2seq paradigm, and randomly select $N_{neg}$ unobserved items within the same domain for each positive sample to form the negative set. We use InfoNCE [48] to optimize ABXI. Given a recommendation representation $h$, we denote the embedding of the corresponding positive sample as $e^+$, and the embedding set of the union of positive and negative samples as $E$. Therefore, the InfoNCE can thus be given as:

$$f(h) = -\log \frac{\exp(h \cdot e^+ / \tau)}{\sum_{e \in E} \exp(h \cdot e / \tau)}, \qquad (11)$$

where $\tau$ denotes the temperature factor. The final loss of sequence can then be obtained as:

$$Loss = \frac{1}{|H_A^{rec}|} \sum_{h \in H_A^{rec}} f(h) + \frac{1}{|H_B^{rec}|} \sum_{h \in H_B^{rec}} f(h). \qquad (12)$$

## 4 EXPERIMENTS

We design our experiments to answer the following research questions:

**RQ1**: How does ABXI perform in comparison to state-of-the-art CDSR models?

**RQ2**: How do the proposed task-guided alignment, projectors, iLoRA, and dLoRA of ABXI benefit its performance?

**RQ3**: What is the impact on ABXI's performance when replacing the proposed iLoRAs or dLoRAs with dense layers?

**RQ4**: How does the choice of rank hyperparameter in iLoRA and dLoRA affect the performance of ABXI?

## 4.1 Datasets

We conduct our experiments on three datasets derived from the Amazon review datasets[1] [40], encompassing six distinct domains: Food-Kitchen (FK), Beauty-Electronics (BE), and Movie-Book (MB). Specifically, FK includes the 'Grocery and Gourmet Food' as A and 'Home and Kitchen' as B; BE comprises 'Beauty' as A and 'Electronics' as B; MB consists of 'Movies and TV' as A and 'Books' as B.

In our preprocessing setup, each review is treated as a user interaction. We retain users who have interacted in both domains, aggregating and reordering their interactions chronologically based on the timestamps. Subsequently, we remove items that have been interacted with fewer than five times among these users. To further

---

[1]https://cseweb.ucsd.edu/~jmcauley/datasets/amazon/links.html

reduce computational load, we limit each user's interaction sequence to the latest 50 interactions, following [21]. This truncation may result in some users no longer meeting the domain-overlapping criteria, necessitating a secondary filtering step to exclude these users.

The statistics of the processed datasets are summarized in Table 1. We evaluate all methods using five different random seeds to ensure the robustness and reproducibility of the results. Performance is measured using Hit Rate (HR) and Normalized Discounted Cumulative Gain (NDCG) [34] at cutoff values K={5, 10}. For single-target models, hyperparameters are selected based on the NDCG@10 score within each domain. In contrast, hyperparameters are optimized for dual-target models based on the aggregate NDCG@10 scores across both domains.

## 4.2 Baselines

We compare ABXI with several baseline models, categorized into four types:

**ST-SDSR** Single-Target Single-Domain Sequential Recommenders: SASRec-1 and BERT4Rec-1.

**DT-SDSR** Dual-Target Single-Domain Sequential Recommender: SASRec-2 and BERT4Rec-2.

**ST-CDSR** Single-Target Cross-Domain Sequential Recommender: CD-SASRec, CD-ASR and MGCL.

**DT-CDSR** Dual-Target Cross-Domain Sequential Recommender: C²DSR and DREAM.

These baseline models are described as follows:

- **SASRec** [21] utilizes the self-attention encoder to generate sequential representations. We implement two versions: ST-SASRec-1 and SASRec-2, corresponding to ST and DT settings, respectively. Specifically, losses in SASRec-2 are calculated separately within each domain and then summed.
- **BERT4Rec** [45] introduce Cloze objectives on top of SASRec. Similarly, we use two versions: BERT4Rec-1 and BERT4Rec-2.
- **CD-SASRec** [5] aggregates the encoded source-domain sequences into the target-domain encoding using two self-attention encoders.
- **CD-ASR** [4] fuses source and target domain sequences encoded by separate self-attention encoders.
- **C²DSR** [9] instantiate difference set of graphical and self-attention encoder to encode cross-domain and domain-specific sequences, leveraging augmentation for contrastive learning.
- **MGCL** [53] integrates graphical and sequential information under different views and strengthens the profiling via user-to-user contrastive learning on views.
- **DREAM** [55] employs separate self-attention encoders for cross-domain and domain-specific sequences, incorporating specific-to-cross knowledge transferring and a similar user-to-user contrastive learning on domains.

## 4.3 Implementation Details

We adopt the leave-one-out strategy commonly used in SR. Specifically, we remove the last two interactions from each user sequence to serve as the ground truths for validation and testing. Evaluation metrics are computed separately for each domain. Besides basic statistics, Table 1 summarizes the number of validation and testing

**Table 1: Statistics of CDSR Datasets.**

| Datasets | FK | | BE | | MB | |
|---|---|---|---|---|---|---|
| | Food (A) | Kitchen (B) | Beauty (A) | Electronics (B) | Movie (A) | Book (B) |
| Users | 7,144 | | 4,474 | | 28,350 | |
| Items | 11,837 | 16,258 | 10,379 | 14,188 | 35,712 | 90,958 |
| Interactions | 83,663 | 89,885 | 50,329 | 63,800 | 347,654 | 403,147 |
| Val. GT | 2,837 | 4,307 | 2,086 | 2,388 | 11,728 | 16,622 |
| Test GT | 2,419 | 4,725 | 1,875 | 2,599 | 10,935 | 17,415 |
| A → B transitions | 30,308 | - | 17,841 | - | 108,318 | - |
| B → A transitions | - | 29,407 | - | 17,888 | - | 105,696 |

**Table 2: Hyperparameters selection.**

| Hyperparameter | Value |
|---|---|
| Embedding dimension $d$ | 256 |
| # self-attention layer | 1 |
| dropout rate | 0.3 |
| # negative sample $N_{neg}$ | 128 |
| Optimizer | AdamW |
| Temperature $\tau$ | 0.75 |
| Max epoch | 500 |
| Warm-up epoch | 5 |
| Learning rate | {1e-3, 1e-4} |
| Weight decay | {5, 2, 1}×{1e1, 1e0, 1e-1, 1e-2, 1e-3}, 0 |
| Learning rate decay | ×0.3162, after 30 stable epochs |
| Random seed | {3407, 0, 1, 2, 3} |

ground truths per domain, as well as the counts of cross-domain item-to-item transitions within all sequences (i.e., A→B and B→A). Hyperparameters not specified elsewhere are listed in Table 2.

Our experimental setup treats CDSR as SDSR with added side domain information. By comparing both types of models under identical conditions, we aim to assess the performance improvements fairly through CDSR. If new CDSR models do not outperform classic SDSR models within this setting, their practical utility may be limited.

### 4.4 Overall Performance Comparison (RQ1)

To evaluate ABXI and address RQ1, we compare our model with state-of-the-art CDSR models and other baseline methods. For each dataset, we designate the domain with better performance metrics as the 'easy' domain and the other as the 'hard' domain. The overall results are reported in Table 3.

Across all datasets, ABXI significantly outperforms all baseline models on all evaluation metrics, with statistical significance (p<0.01). This improvement is evident in both domains, indicating that a single sequence model augmented with auxiliary modules is sufficient to capture both cross-domain and domain-specific knowledge.

In contrast, other DT-CDSR models do not demonstrate substantial improvements over their SDSR counterparts. Only in the Movie domain do we observe that DREAM outperforms the SDSR models.

This can be attributed to the prediction mismatches in timestamp-guided alignment within these DT-CDSR models, causing them to make predictions in incorrect domains. Conversely, ST- and DT-SDSR models do not encounter such issues, as they do not require domain-specific sequences.

Furthermore, the ST-CDSR models demonstrate notably poor performance compared to all other types of models. This underperformance stems from their training from scratch without supervision in the source domains, which hinders the extraction of domain-invariant information. Without supervision from the source domain, these models cannot effectively distinguish domain-invariant knowledge from domain-specific knowledge. This results in source-specific information, which should have been filtered out, being introduced into the target domain.

### 4.5 Ablation Studies (RQ2)

To further investigate the contribution of each proposed component, we designed five ablation variants: $V_1$ removes all dLoRAs; $V_2$ removes all projectors; $V_3$ removes all iLoRAs; $V_4$ removes all dLoRAs, iLoRAs and projectors; $V_5$ replaces the task-guided alignment with the timestamp-guided alignment. The results of these ablation studies are reported in the middle section of Table 4.

Compared to all ablation variants, ABXI achieves the best performance. Notably, we have the following observations.

Projectors contribute the most among all trainable modules. Due to their substantial number of learnable parameters, projectors provide sufficient capacity to transform encoded sequential representations into recommendation representations.

The performance gaps between $V_1$, $V_3$ and ABXI, as well as between $V_2$ and $V_4$, indicate that iLoRA and dLoRA effectively enhance performance with few additional parameters.

The performance improvements of ABXI over $V_1$ and $V_3$, as well as of $V_2$ over $V_4$, demonstrate that iLoRA and dLoRA effectively enhance performance with a small number of additional parameters.

Among all variants, reverting to timestamp-guided alignment in $V_5$ results in the most significant performance degradation. Since ABXI does not employ specialized domain-specific ground truths under timestamp-guided alignment, it suffers severely from the prediction mismatch issue.

**Table 3: Recommendation performance (RQ1). The best and the runner-up are highlighted in bold and underlined respectively.**

| Type | Methods | Food | | | | Kitchen | | | |
|---|---|---|---|---|---|---|---|---|---|
| | | HR@5 | HR@10 | NDCG@5 | NDCG@10 | HR@5 | HR@10 | NDCG@5 | NDCG@10 |
| ST | SASRec-1 | $0.1867_{\pm0.0041}$ | $0.2530_{\pm0.0037}$ | $0.1309_{\pm0.0033}$ | $0.1524_{\pm0.0032}$ | $0.1213_{\pm0.0029}$ | $0.1786_{\pm0.0019}$ | $0.0822_{\pm0.0014}$ | $0.1006_{\pm0.0014}$ |
| | BERT4Rec-1 | $0.1812_{\pm0.0030}$ | $0.2482_{\pm0.0038}$ | $0.1244_{\pm0.0021}$ | $0.1461_{\pm0.0022}$ | $0.1099_{\pm0.0042}$ | $0.1652_{\pm0.0059}$ | $0.0736_{\pm0.0032}$ | $0.0914_{\pm0.0036}$ |
| | CD-SASRec | $0.1641_{\pm0.0163}$ | $0.2277_{\pm0.0135}$ | $0.1119_{\pm0.0131}$ | $0.1326_{\pm0.0120}$ | $0.0993_{\pm0.0060}$ | $0.1594_{\pm0.0082}$ | $0.0645_{\pm0.0044}$ | $0.0837_{\pm0.0045}$ |
| | CD-ASR | $0.1892_{\pm0.0022}$ | $0.2590_{\pm0.0050}$ | $0.1287_{\pm0.0025}$ | $0.1513_{\pm0.0033}$ | $0.1146_{\pm0.0016}$ | $0.1727_{\pm0.0023}$ | $0.0760_{\pm0.0017}$ | $0.0946_{\pm0.0018}$ |
| | MGCL | $0.1753_{\pm0.0061}$ | $0.2465_{\pm0.0052}$ | $0.1191_{\pm0.0026}$ | $0.1423_{\pm0.0025}$ | $0.1173_{\pm0.0073}$ | $0.1803_{\pm0.0111}$ | $0.0782_{\pm0.0046}$ | $0.0985_{\pm0.0059}$ |
| DT | SASRec-2 | $0.2269_{\pm0.0021}$ | $0.2857_{\pm0.0017}$ | $0.1568_{\pm0.0018}$ | $0.1760_{\pm0.0024}$ | $0.1458_{\pm0.0030}$ | $0.2105_{\pm0.0031}$ | $0.0988_{\pm0.0018}$ | $0.1196_{\pm0.0018}$ |
| | BERT4Rec-2 | $0.2157_{\pm0.0046}$ | $0.2827_{\pm0.0042}$ | $0.1475_{\pm0.0054}$ | $0.1693_{\pm0.0041}$ | $0.1366_{\pm0.0051}$ | $0.2047_{\pm0.0050}$ | $0.0904_{\pm0.0039}$ | $0.1123_{\pm0.0038}$ |
| | C$^2$DSR | $0.2071_{\pm0.0074}$ | $0.2669_{\pm0.0053}$ | $0.1453_{\pm0.0028}$ | $0.1647_{\pm0.0029}$ | $0.1244_{\pm0.0037}$ | $0.1841_{\pm0.0069}$ | $0.0850_{\pm0.0024}$ | $0.1043_{\pm0.0033}$ |
| | DREAM | $0.1947_{\pm0.0041}$ | $0.2521_{\pm0.0090}$ | $0.1369_{\pm0.0022}$ | $0.1555_{\pm0.0038}$ | $0.1244_{\pm0.0039}$ | $0.1790_{\pm0.0060}$ | $0.0830_{\pm0.0031}$ | $0.1007_{\pm0.0037}$ |
| | **ABXI** | **$0.2499_{\pm0.0037}$** | **$0.3187_{\pm0.0038}$** | **$0.1752_{\pm0.0035}$** | **$0.1977_{\pm0.0033}$** | **$0.1757_{\pm0.0039}$** | **$0.2439_{\pm0.0029}$** | **$0.1205_{\pm0.0017}$** | **$0.1425_{\pm0.0009}$** |

| Type | Methods | Beauty | | | | Electronics | | | |
|---|---|---|---|---|---|---|---|---|---|
| | | HR@5 | HR@10 | NDCG@5 | NDCG@10 | HR@5 | HR@10 | NDCG@5 | NDCG@10 |
| ST | SASRec-1 | $0.1616_{\pm0.0021}$ | $0.2286_{\pm0.0041}$ | $0.1130_{\pm0.0020}$ | $0.1346_{\pm0.0027}$ | $0.1172_{\pm0.0026}$ | $0.1711_{\pm0.0031}$ | $0.0803_{\pm0.0009}$ | $0.0976_{\pm0.0010}$ |
| | BERT4Rec-1 | $0.1648_{\pm0.0031}$ | $0.2354_{\pm0.0037}$ | $0.1141_{\pm0.0034}$ | $0.1368_{\pm0.0037}$ | $0.1307_{\pm0.0020}$ | $0.1870_{\pm0.0016}$ | $0.0901_{\pm0.0016}$ | $0.1082_{\pm0.0013}$ |
| | CD-SASRec | $0.1303_{\pm0.0104}$ | $0.2016_{\pm0.0128}$ | $0.0887_{\pm0.0081}$ | $0.1116_{\pm0.0083}$ | $0.1131_{\pm0.0072}$ | $0.1645_{\pm0.0104}$ | $0.0766_{\pm0.0050}$ | $0.0932_{\pm0.0057}$ |
| | CD-ASR | $0.1587_{\pm0.0042}$ | $0.2389_{\pm0.0042}$ | $0.1082_{\pm0.0035}$ | $0.1341_{\pm0.0033}$ | $0.1258_{\pm0.0029}$ | $0.1842_{\pm0.0020}$ | $0.0856_{\pm0.0027}$ | $0.1044_{\pm0.0023}$ |
| | MGCL | $0.1400_{\pm0.0076}$ | $0.2096_{\pm0.0090}$ | $0.0952_{\pm0.0056}$ | $0.1176_{\pm0.0060}$ | $0.1272_{\pm0.0033}$ | $0.1828_{\pm0.0055}$ | $0.0863_{\pm0.0027}$ | $0.1043_{\pm0.0031}$ |
| DT | SASRec-2 | $0.2371_{\pm0.0062}$ | $0.3210_{\pm0.0037}$ | $0.1678_{\pm0.0028}$ | $0.1949_{\pm0.0020}$ | $0.1353_{\pm0.0032}$ | $0.2041_{\pm0.0038}$ | $0.0901_{\pm0.0027}$ | $0.1124_{\pm0.0021}$ |
| | BERT4Rec-2 | $0.1810_{\pm0.0060}$ | $0.2794_{\pm0.0063}$ | $0.1228_{\pm0.0046}$ | $0.1545_{\pm0.0048}$ | $0.1467_{\pm0.0029}$ | $0.2148_{\pm0.0026}$ | $0.1002_{\pm0.0025}$ | $0.1221_{\pm0.0028}$ |
| | C$^2$DSR | $0.1927_{\pm0.0090}$ | $0.2785_{\pm0.0122}$ | $0.1310_{\pm0.0051}$ | $0.1588_{\pm0.0058}$ | $0.1361_{\pm0.0052}$ | $0.1905_{\pm0.0060}$ | $0.0930_{\pm0.0033}$ | $0.1105_{\pm0.0030}$ |
| | DREAM | $0.2007_{\pm0.0077}$ | $0.2918_{\pm0.0067}$ | $0.1362_{\pm0.0045}$ | $0.1656_{\pm0.0035}$ | $0.1164_{\pm0.0065}$ | $0.1733_{\pm0.0071}$ | $0.0794_{\pm0.0043}$ | $0.0977_{\pm0.0043}$ |
| | **ABXI** | **$0.2722_{\pm0.0024}$** | **$0.3722_{\pm0.0025}$** | **$0.1862_{\pm0.0030}$** | **$0.2186_{\pm0.0025}$** | **$0.1642_{\pm0.0044}$** | **$0.2389_{\pm0.0061}$** | **$0.1138_{\pm0.0036}$** | **$0.1377_{\pm0.0041}$** |

| Type | Methods | Movie | | | | Book | | | |
|---|---|---|---|---|---|---|---|---|---|
| | | HR@5 | HR@10 | NDCG@5 | NDCG@10 | HR@5 | HR@10 | NDCG@5 | NDCG@10 |
| ST | SASRec-1 | $0.2258_{\pm0.0031}$ | $0.2961_{\pm0.0037}$ | $0.1647_{\pm0.0025}$ | $0.1874_{\pm0.0027}$ | $0.1357_{\pm0.0029}$ | $0.1789_{\pm0.0033}$ | $0.1007_{\pm0.0022}$ | $0.1147_{\pm0.0023}$ |
| | BERT4Rec-1 | $0.2155_{\pm0.0021}$ | $0.2907_{\pm0.0019}$ | $0.1546_{\pm0.0016}$ | $0.1789_{\pm0.0015}$ | $0.1429_{\pm0.0020}$ | $0.1874_{\pm0.0025}$ | $0.1069_{\pm0.0009}$ | $0.1212_{\pm0.0011}$ |
| | CD-SASRec | $0.2187_{\pm0.0065}$ | $0.2944_{\pm0.0074}$ | $0.1555_{\pm0.0044}$ | $0.1800_{\pm0.0047}$ | $0.1413_{\pm0.0081}$ | $0.1829_{\pm0.0082}$ | $0.1045_{\pm0.0065}$ | $0.1179_{\pm0.0064}$ |
| | CD-ASR | $0.2088_{\pm0.0011}$ | $0.2819_{\pm0.0015}$ | $0.1492_{\pm0.0015}$ | $0.1728_{\pm0.0017}$ | $0.1294_{\pm0.0015}$ | $0.1742_{\pm0.0014}$ | $0.0925_{\pm0.0014}$ | $0.1070_{\pm0.0013}$ |
| | MGCL | $0.1980_{\pm0.0084}$ | $0.2662_{\pm0.0078}$ | $0.1416_{\pm0.0083}$ | $0.1636_{\pm0.0080}$ | $0.1163_{\pm0.0011}$ | $0.1533_{\pm0.0016}$ | $0.0866_{\pm0.0009}$ | $0.0985_{\pm0.0010}$ |
| DT | SASRec-2 | $0.2150_{\pm0.0026}$ | $0.2837_{\pm0.0013}$ | $0.1557_{\pm0.0013}$ | $0.1779_{\pm0.0011}$ | $0.1251_{\pm0.0014}$ | $0.1671_{\pm0.0016}$ | $0.0931_{\pm0.0011}$ | $0.1066_{\pm0.0013}$ |
| | BERT4Rec-2 | $0.2064_{\pm0.0022}$ | $0.2790_{\pm0.0028}$ | $0.1476_{\pm0.0014}$ | $0.1710_{\pm0.0017}$ | $0.1399_{\pm0.0011}$ | $0.1854_{\pm0.0016}$ | $0.1038_{\pm0.0004}$ | $0.1185_{\pm0.0005}$ |
| | C$^2$DSR | $0.2135_{\pm0.0049}$ | $0.2813_{\pm0.0042}$ | $0.1551_{\pm0.0036}$ | $0.1770_{\pm0.0034}$ | $0.1155_{\pm0.0053}$ | $0.1538_{\pm0.0063}$ | $0.0863_{\pm0.0037}$ | $0.0987_{\pm0.0039}$ |
| | DREAM | $0.2308_{\pm0.0073}$ | $0.3060_{\pm0.0084}$ | $0.1665_{\pm0.0058}$ | $0.1908_{\pm0.0062}$ | $0.1299_{\pm0.0058}$ | $0.1766_{\pm0.0076}$ | $0.0938_{\pm0.0043}$ | $0.1089_{\pm0.0047}$ |
| | **ABXI** | **$0.2790_{\pm0.0030}$** | **$0.3590_{\pm0.0015}$** | **$0.2078_{\pm0.0016}$** | **$0.2335_{\pm0.0014}$** | **$0.1901_{\pm0.0015}$** | **$0.2472_{\pm0.0015}$** | **$0.1429_{\pm0.0009}$** | **$0.1613_{\pm0.0007}$** |

**Table 4: Ablation studies in NDCG@10. (RQ2, 3)**

| | iLoRA | Proj | dLoRA | Alignment | Food | Kitchen | Beauty | Electronics |
|---|---|---|---|---|---|---|---|---|
| ABXI | ✓ | ✓ | ✓ | task | **$0.1977_{\pm0.0033}$** | **$0.1425_{\pm0.0009}$** | $0.2186_{\pm0.0025}$ | **$0.1377_{\pm0.0041}$** |
| V$_1$ | ✓ | ✓ | - | task | **$0.1977_{\pm0.0026}$** | $0.1417_{\pm0.0017}$ | $0.2151_{\pm0.0020}$ | $0.1353_{\pm0.0014}$ |
| V$_2$ | ✓ | - | ✓ | task | $0.1938_{\pm0.0013}$ | $0.1381_{\pm0.0015}$ | $0.2092_{\pm0.0015}$ | $0.1345_{\pm0.0039}$ |
| V$_3$ | - | ✓ | ✓ | task | $0.1970_{\pm0.0027}$ | $0.1402_{\pm0.0011}$ | $0.2173_{\pm0.0028}$ | $0.1349_{\pm0.0026}$ |
| V$_4$ | - | - | - | task | $0.1923_{\pm0.0034}$ | $0.1327_{\pm0.0018}$ | $0.2062_{\pm0.0024}$ | $0.1366_{\pm0.0026}$ |
| V$_5$ | ✓ | ✓ | ✓ | timestamp | $0.1753_{\pm0.0028}$ | $0.1305_{\pm0.0032}$ | $0.2011_{\pm0.0082}$ | $0.1198_{\pm0.0017}$ |
| ABXI-i3 | 3×Proj | ✓ | ✓ | task | $0.1954_{\pm0.0025}$ | $0.1415_{\pm0.0022}$ | $0.2185_{\pm0.0022}$ | $0.1348_{\pm0.0032}$ |
| ABXI-i2 | 2×Proj | ✓ | ✓ | task | $0.1953_{\pm0.0037}$ | $0.1413_{\pm0.0020}$ | **$0.2191_{\pm0.0042}$** | $0.1366_{\pm0.0034}$ |
| ABXI-d | ✓ | ✓ | 3×encoders | task | $0.1976_{\pm0.0016}$ | $0.1405_{\pm0.0018}$ | $0.2114_{\pm0.0042}$ | $0.1329_{\pm0.0031}$ |

## 4.6 Effectiveness of LoRA (RQ3)

LoRA is renowned for its efficient adaptation. However, in the context of non-LLM CDSR, we need to determine whether this efficiency translates into effectiveness. Therefore, we design the following three variants: **ABXI-i3**: replace iLoRA$_A$ and iLoRA$_B$ with X-to-A and X-to-A projectors; **ABXI-i2**: remove the Proj$_i$ in ABXI-i3; **ABXI-d**: replace the shared encoder with dLoRAs with three self-attention encoders for domains $X$, $A$ and $B$, respectively. The results are reported in the bottom section of Table 4.

Among them, ABXI-i2 achieves performance closely matching that of ABXI. This suggests redundancy between the X-to-A and X-to-A projectors, which indicates that they can be functionally replaced by the combination of the Proj$_i$ and two iLoRA modules.

ABXI-i3 utilizes the shared Proj$_i$ along with specialized X-to-A and X-to-A projectors. However, this configuration does not yield significant performance improvements. The introduction of these projectors substantially increases the capacity for cross-to-specific transformations, which diminishes ABXI's reliance on the Proj$_i$ and impedes its ability to extract domain-invariant knowledge.

ABXI-d degrades performance by instantiating three encoders for each domain instead of the shared one with dLoRAs. We attribute this degradation to our task-guided alignment. Since we unify the downstream recommendation tasks, we do not need to employ additional domain-specific ground truths like C$^2$DSR [9] and DREAM [55]. Therefore, these domain-specific encoders cannot achieve the same performance as they do in those baselines.

## 4.7 Rank Analysis in LoRA (RQ4)

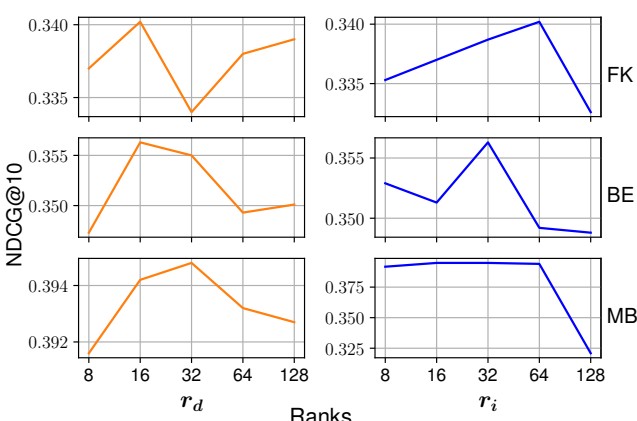

**Figure 4: Performance of all domains under different values of ranks of dLoRA ($r_d$) and iLoRA ($r_i$) in NDCG@10 (RQ4).**

This subsection investigates the impact of different rank values $r_d$ for dLoRA and $r_i$ for iLoRA on the ABXI's performance. We vary these ranks over {8, 16, 32, 64, 128} and present the results in Figure 4.

From Figure 4, we observe a general tendency across all six domains: both large and small ranks tend to yield poorer results, although some fluctuations are evident. This behavior is due to the limited capacity of LoRA modules within the one-stage training

framework of ABXI. To achieve optimal performance, we select ($r_d = 16, r_i = 64$) for FK, ($r_d = 16, r_i = 32$) for BE, and ($r_d = 32, r_i = 32$) for MB.

## 5 CONCLUSION

In this paper, we propose ABXI, a novel CDSR model. Specifically, ABXI addresses the prediction mismatch issue by integrating task-guided alignment to unify cross-domain and domain-specific optimizations. ABXI leverages domain projectors equipped with invariant LoRA modules to extract and adapt domain-invariant interests, enabling efficient and effective knowledge transfer Furthermore, ABXI employs a single shared encoder with domain LoRA to conduct efficient encoding. Extensive experimental results on three datasets demonstrated that ABXI significantly outperforms state-of-the-art CDSR models by a large margin. Ablation studies confirmed the effectiveness of each component, highlighting the importance of task-guided alignment and invariant interest extraction. For future work, we aim to further explore domain-invariant dynamics to achieve more accurate disentanglement.

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
