# OpenReview forum: "ABXI: Invariant Interest Adaptation for Task-Guided Cross-Domain Sequential Recommendation"
_ACM.org/TheWebConf/2025/Conference — WWW 2025 Poster_

### Official Review · Reviewer_fvhY · 2024-11-18

**Novelty:** 5
**Technical Quality:** 6

**Review:**

This paper proposes a novel cross-domain sequential recommendation system framework called ABXI. The ABXI model stresses the prediction mismatch problem in existing CDSR models with task-guided alignment and includes two types of LoRA to learn domain-specific and domain-invariant information correspondingly.
Paper Strengths:
1.	The presentation of this paper is clear and easy to follow. The paper includes sufficient explanations of different parts of the model design, which makes the paper solid and convincing.
2.	This paper clearly illustrates the prediction mismatch problem and the task-guided alignment with a figure.
3.	Extensive experiments are included to demonstrate the effectiveness of the ABXI model. A series of ablation studies also show the effectiveness of the LoRAs and the task-guided alignment.
Paper Weaknesses:
1.	The writers are encouraged to include more theoretical analysis, including a time complexity analysis.
2.	The presentation can be further improved by correcting some mistakes. (For more details, please refer to the “Questions” part.)
In general, in this paper, the idea of introducing LoRAs to the field of cross-domain recommendation is novel to some extent, and the presentation is solid with high technical quality.

**Questions:**

1.	In Figure 2(b), the gt_B is mismatched with seq_B. Should the items in gt_B be shifted one position right?
2.	In the last paragraph of Section 3.3, “where N the total number of items,” is there an “is” missing here? In addition, is the definition of the position embedding table duplicated?
3.	In Section 3.4, the writers argue that LoRA is not only compatible with multi-stage training but can also be effective in single-stage training. It would be more convincing if the writers could further explain this argument.

**Reviewer Confidence:**

3: The reviewer is confident but not certain that the evaluation is correct

**Scope:**

4: The work is relevant to the Web and to the track, and is of broad interest to the community

---

### Official Review · Reviewer_A3dB · 2024-11-24

**Novelty:** 3
**Technical Quality:** 4

**Review:**

$\textbf{Summary}$

The authors consider that the main challenges for cross-domain sequential recommendation include shared knowledge exaction and transfer from different domains and cross-domain and domain-specific sequence alignment. However, they figure out that most existing works directly transfer unfiltered cross-domain knowledge and face the problem of prediction mismatch. As a result, the authors propose ABXI in this paper, which designs task-guided alignment to avoid prediction mismatch. In addition, two types of LoRAs, i.e. dLoRA and iLoRA, are leveraged to extract the representations of domain-specific and cross-domain knowledge.

$\textbf{Strengths}$

S1: The authors design several components to tackle the challenges of CDSR.

S2: Codes are provided for easy reproducibility.

$\textbf{Weaknesses}$

W1: In the ablation study, when removing the task-guided alignment, ABIX sometimes fails to outperform traditional methods (On the food dataset, it is lower than SASRec-2. While on the electronics, it is lower than BERT4Rec-2). Therefore, the effect of other components such as LoRAs seems to be slight. I think more results of the ablation study could be presented to verify their effectiveness.

W2: Compared with existing CDSR works, there seems to be no obvious innovation or superiority.

W3: What’s the effect of the temperature $\tau$? Since InfoNCE is the only training goal, it may have some effect on the performance of ABXI.

**Questions:**

Please see the weaknesses in the above part.

**Reviewer Confidence:**

3: The reviewer is confident but not certain that the evaluation is correct

**Scope:**

4: The work is relevant to the Web and to the track, and is of broad interest to the community

---

### Official Review · Reviewer_uq4X · 2024-11-29

**Novelty:** 4
**Technical Quality:** 5

**Review:**

The authors propose the A-B-Cross-to-Invariant Learning Recommender (ABXI). Specifically, ABXI addresses the prediction mismatch issue by integrating task-guided alignment to unify cross-domain and domain-specific optimizations. Extensive experimental results on three
datasets demonstrated that ABXI significantly outperforms state-of-the-art CDSR models by a large margin. The method exhibits certain originality in the field of cross-domain recommendation, particularly with the integration of task-guided alignment and LoRA technology. The paper is well-structured, with clear descriptions of the approach and effective illustrations, demonstrating its potential to significantly improve the accuracy of recommendation systems.

**Questions:**

1、The task-guided alignment method discussed in the paper appears to be designed primarily for two domains. This approach aligns domain-specific and cross-domain sequences to avoid prediction mismatches. However, when extending to more than two domains, does ABXI still perform effectively, or does it face challenges? For instance, in the case of three or more domains (e.g., books, movies, music), can ABXI maintain its effectiveness, or does domain interference start to degrade performance?
2、ABXI aims to extract domain-invariant interests from cross-domain sequences. However, if a user's interests in one domain significantly differ from their interests in another domain—e.g., a user enjoys science fiction books but has little interest in comedy movies—how does ABXI ensure that it extracts relevant, domain-invariant interests without introducing noise from irrelevant domains?
3、 The paper mentions that LoRA modules (dLoRA and iLoRA) effectively adjust the model's behavior and representations to enhance cross-domain recommendation performance. However, do the LoRA modules introduce any computational overhead? When scaling to large datasets or multiple domains, do the LoRAs' computational costs impact the model’s efficiency?

**Reviewer Confidence:**

3: The reviewer is confident but not certain that the evaluation is correct

**Scope:**

4: The work is relevant to the Web and to the track, and is of broad interest to the community

---

### Official Review · Reviewer_WMgL · 2024-12-02

**Novelty:** 4
**Technical Quality:** 4

**Review:**

The paper addresses the problem of Cross-Domain Sequential Recommendation (CDSR), focusing on transferring knowledge across domains to counteract data sparsity. Most previous approaches have been limited in that they directly transfer cross-domain knowledge without properly distinguishing between domain-specific and domain-invariant components. The paper proposes a method for invariant interest adaptation, where shared knowledge between domains is adaptively integrated into domain-specific models. The core contribution is to improve the alignment between domain-specific and cross-domain sequences, addressing mismatches that arise from using timestamps for alignment, which may cause prediction errors.
Pos:
1. The paper identifies specific challenges with existing CDSR approaches, especially the misalignment caused by timestamp-based sequence alignment. By addressing these gaps, the paper makes a valuable contribution to improving the accuracy and reliability of cross-domain recommendations.
2. The focus on handling data sparsity through cross-domain transfer makes the proposed approach highly relevant for real-world recommendation systems, where limited data from individual domains is a major issue. This has the potential for wide application in fields like e-commerce, entertainment, and personalized services
3. Extensive experiment are conducted to demonstrate the effectiveness of the proposed methods.
Cons:
1. The paper lacks in-depth experimental evaluation to demonstrate its superiority over existing methods. The authors should provide more results according to the accuracy, efficiency, and scalability. Besides, statistical significance tests should be provided to support claims.
2. The rationality behind this method should be detailed, currently, the description seems naive and lacks in-depth analysis.
3. The paper could benefit from concrete examples or case studies showing how the method performs across various domains. Currently, the approach is presented abstractly, which might make it harder for the reader to envision its practical application.

**Questions:**

1. The paper mentions alignment based on timestamps, which can lead to prediction mismatches. Could you explain how your model overcomes this issue in more detail? What methods or metrics are used to ensure correct alignment when timestamps are not sufficient?
2. How exactly do you define and extract the "domain-invariant" components in your approach? Is this done through some form of feature selection, clustering, or other means? A more detailed explanation of the process would help in evaluating the robustness of your method.

**Reviewer Confidence:**

3: The reviewer is confident but not certain that the evaluation is correct

**Scope:**

4: The work is relevant to the Web and to the track, and is of broad interest to the community

---

### Official Review · Reviewer_ZPtB · 2024-12-02

**Novelty:** 5
**Technical Quality:** 4

**Review:**

**Quality**

The paper introduces ABXI, a novel approach to Cross-Domain Sequential Recommendation (CDSR) that addresses key challenges such as domain-invariant knowledge extraction and prediction mismatch. This work is particularly noteworthy for its innovative approach to blending domain-specific sequences into cross-domain sequences, which serves as a bridge for mutual enhancement between domains. The experimental results demonstrate ABXI's effectiveness, with significant improvements over state-of-the-art methods in HR@10 and NDCG@10 across various datasets. This suggests that ABXI is capable of effectively transferring knowledge across domains, thereby enriching user profiles and countering data sparsity.

**Clarity**

The paper is well-structured, with clear definitions and methodology explanations that guide the reader through the complex problem of cross-domain sequential recommendation. The use of diagrams is effective in illustrating complex concepts such as task-guided alignment and the role of invariant LoRAs in the model. However, there is room for improvement in explaining certain technical terms, which could benefit from more context or background for readers who are not experts in the field. Additionally, while the introduction frames the problem well, the conclusion could be strengthened by discussing potential limitations and future work, providing a more comprehensive overview of the implications and broader impact of the research.

**Originality**

The use of task-guided alignment to resolve prediction mismatch and the integration of Low-Rank Adaptation (LoRA) for domain-specific and invariant modeling are innovative contributions to the field of recommender systems. These approaches stand out as creative solutions to the challenges of extracting and transferring knowledge across domains. The paper also extends the application of LoRA from fine-tuning to single-stage training, which is a novel contribution that could have broader implications for how we approach domain adaptation in recommender systems.

**Significance**

The work is significant as it offers practical solutions for addressing data sparsity in cross-domain settings, a common issue in modern recommender systems. By achieving substantial performance improvements, ABXI establishes itself as a valuable contribution to both academia and industry.

**Pros:**

- Innovative approach to resolving prediction mismatch in CDSR.
- Effective use of LoRA for domain adaptation.
- Significant performance improvements over state-of-the-art methods.
- Comprehensive experiment validating the model design.

**Cons:**

- Further analysis on the effectiveness of LoRA compared to full-parameter matrices is needed.
- More details on the dataset processing and its applicability to other datasets are required.
- An analysis of resource consumption and computational complexity is lacking.

**Questions:**

**Questions:**

1. **LoRA Effectiveness and Significance:**
   - In the ablation study, the performance difference between LoRA and projectors was minimal, with around 0.002 improvement at the thousandth place level, and in some domains, such as `beauty`, the projector performed even better. Additionally, considering the more substantial improvement comes from the alignment method (with a 0.02 point of improvement), can the authors discuss the relative impact of LoRA compared to alignment strategies on the model's performance?
   - Have the authors experimented with replacing dLoRA with projectors, and if so, what were the results of such experiments?
   - Given that current literature on large language models (LLMs) suggests that full-parameter tuning generally outperforms LoRA, and LoRA often used as a way of improving finetuning efficiency and reducing resource consumptions\[1\], has the authors considered whether using the original full-parameter matrices might yield better performance, especially when computational resources are not a constraint? Have the authors conducted any related experiments to compare LoRA with full-parameter tuning?
2. **Dataset Processing and Applicability to Other Datasets:**
   - How is the secondary filtering step for the Amazon dataset handled, and how does this process scale to larger datasets or other cross-domain datasets?
   - Can the secondary filtering step be applied to scenarios involving more than two domains?
3. **Resource Consumption and Complexity Analysis:**
   - Could the authors provide an analysis of the resource consumption and computational complexity associated with the ABXI model?
4. **Code Repository Access:**
   - The provided link to the code repository (https://anonymous.4open.science/status/ABXI-WWW25-1D04) is not accessible. Could the authors provide an updated link or alternative means to access the code?

[1] Hu, Edward J., et al. "Lora: Low-rank adaptation of large language models." arXiv preprint arXiv:2106.09685 (2021).

**Reviewer Confidence:**

3: The reviewer is confident but not certain that the evaluation is correct

**Scope:**

4: The work is relevant to the Web and to the track, and is of broad interest to the community